# Whole Transcriptome Analysis of Differentially Expressed Genes in Cultured Nile Tilapia (*O. niloticus*) Subjected to Chronic Stress Reveals Signaling Pathways Associated with Depressed Growth

**DOI:** 10.3390/genes14040795

**Published:** 2023-03-25

**Authors:** John Gitau Mwaura, Clabe Wekesa, Philip A. Ogutu, Patrick Okoth

**Affiliations:** 1Department of Biological Sciences, School of Natural and Applied Sciences, Masinde Muliro University of Science and Technology, Kakamega P.O. Box 190-50100, Kenya; 2Department of Fisheries, County Government of Kakamega, Kakamega P.O. Box 586-50100, Kenya

**Keywords:** chronic stress, differentially expressed genes, ammonia, stocking density, MAP-K

## Abstract

Chronic stress is a serious threat to aquaculture as it lowers fish growth performance and compromises fish welfare. The exact mechanism by which growth is retarded is, however, not clearly understood. This study sought to elucidate the gene expression profiles associated with chronic stress in cultured Nile tilapia (*Oreochromis niloticus*) reared for 70 days at different ammonia concentrations and stocking densities. Fish in the treatment groups showed negative growth, while the controls showed positive allometric growth. The specific condition factor (K_n_) ranged from 1.17 for the controls to 0.93 for the ammonia and 0.91 for the stocking density treatments. RNA was extracted from muscle tissue using TRIzol followed by library construction and Illumina sequencing. Comparative transcriptome analysis revealed 209 differentially expressed genes (DEGs) (156 up- and 53 down-regulated) in the ammonia and 252 DEGs (175 up- and 77 down-regulated) in the stocking density treatment. In both treatments, 24 and 17 common DEGs were up- and down-regulated, respectively. DEGs were significantly enriched in six pathways associated with muscle activity, energy mobilization and immunity. The heightened muscular activity consumes energy which would otherwise have been utilized for growth. These results bring to fore the molecular mechanisms underlying chronic stress’ suppression of growth in cultured Nile tilapia.

## 1. Introduction

Fish is a crucial dietary component that accounts for 17% of animal protein and 7% of all the proteins consumed globally [1]. Modern advances in fish genetics, nutrient-rich diets, culture methods, and management techniques have made aquaculture one of the world’s fastest-growing food production industries [2]. Production of tilapia has quadrupled in the last decade, making it the second most farmed fish worldwide [3]. The goal of aquaculture is to enhance production and productivity, which therefore hints at a possible form of raising process intervention [4]. Some of the factors that are highly manipulated include stocking densities, feeds, feeding and genetics [2]. According to projections, the global fish supply will rise from 179 million tons in 2018 to 204 million tons in 2030. Aquaculture production is anticipated to reach 109 million tons this year with the fastest growth expected for tilapia and shrimps [5,6].

Fish get stressed when their environment’s physical and chemical characteristics become unfavorable, e.g., extremes of temperature, pH, salinity, oxygen, ammonia, turbidity and water hardness. Improper fish handling techniques, harassment from other fish and predators, overcrowding, and inadequate nutrition are significant stressors for fish in ponds [7,8]. Exposure of fish to such conditions has been shown to trigger acute or chronic stress reactions based on the degree and length of exposure. Stress is a physiological response of an organism to a threatening situation, whether real or imaginary. Stressors can be intrinsic or extrinsic stimuli that threaten to disturb the dynamic equilibrium of an animal organization called homeostasis [9]. Stress directly influences productivity, and its management is paramount if a farmer is to increase the profitability of a fish farming venture. Furthermore, it has been shown that a stressed fish can influence the stress levels of conspecifics [10], hence the need to maintain a stress-free environment as possible continually.

Ammonia is one of the naturally occurring chemical stressors in aquaculture systems and is one of the primary causes of mass fish mortality. The chief sources of ammonia in fish ponds are ammonification of organic matter such as uneaten fish feeds and other decomposing organic matter, fish fecal matter and fertilizers. Ammonia is a vertebrate’s principal end product of protein catabolism [11]. In freshwater fish, the principal end product of nitrogen metabolism is ammonia [12]. The undigested nitrogenous waste in fish fecal matter, together with the uneaten feeds, tends to settle at the pond bottom and slowly undergo decomposition, producing more ammonia [13]. Ammonia is toxic to fish and other vertebrates and has been shown to cause convulsions, coma and eventually death. In aquaculture systems, ammonia exists in two forms that are in a dynamic equilibrium. The unionized form of ammonia (UIA) is more toxic due to its ability to diffuse through the gills into the fish’s body. The toxicity of ammonia begins at as low as 0.05 mg/L, with fish beginning to die at 2.0 mg/L of UIA. Dissolved oxygen and pH have been shown to directly influence ammonia’s toxicity in aquaculture systems. When the temperature and pH are high, toxicity due to ammonia increases [14,15].

Stressed fish produce more ammonia and are more sensitive to ammonia toxicity from the external environment. Ammonia at concentrations above 0.3 mg/L has been shown to cause death to some freshwater fish. This toxicity may be associated with the displacement of potassium ions by ammonium, resulting in depolarization of neurons leading to activation of a glutamate receptor. Once activated, the receptor allows an influx of excess calcium ions into the CNS, which causes cell death. High environmental pH, especially when the buffering capacity of the pond water is low [16], reduces the gradient for NH_3_ diffusion, leading to a build-up of ammonia inside the fish [17,18].

Stocking density is an important component in influencing the productivity of fish aquaculture systems. Crowding is regarded as one of the most powerful stressors influencing fish physiology and, consequently, the state of well-being in aquaculture. Tilapia’s association behavior changes from antagonistic (hostile) to shoaling due to high stocking density [19,20,21] and their stress coping mechanisms shift from proactive to reactive [22]. Many farm management practices can also induce stress in fish, such as sampling, handling, exposure to air, grading and transportation [23] and thus proper management is mandatory to ensure profitability of a fish farming venture.

Fish growth, development, resistance to disease, behavior, and reproduction are all severely impacted by prolonged stress exposure [24]. Stress response involves the reorganization of the fish’s energy budget, immune system and endocrine mechanisms to cope with the environmental challenge [25]. When confronted with a stressful situation, fish will respond by activating two hormonal axes: the hypothalamo–pituitary–adrenal/interrenal axis (HPA/I) and the sympatho-chromaffin (SC) axis. These responses aim at enabling the fish to escape or overcome the stressor/stressful situation. The SC axis is activated, which leads to enhanced ventilation, which is mediated by an increase in heart rate, the heart stroke volume, and the rate at which huge volumes of blood are sent to the gills and muscles, supplying oxygen and glucose to vital muscles for the escape [26]. The HPI axis, on the other hand, is involved in the reallocation of resources to increase the supply of glucose in the blood by increasing catabolic pathways and depressing other energetically costly processes such as immune responses and reproduction. The HPI axis activation culminates in the release of glucocorticoids from internal cells located in the head kidney [27], leading to production of corticosteroids, primarily cortisol [24,28].

Cortisol also causes chromaffin cells and the endings of adrenergic nerves to increase the release of catecholamines (CAs) (epinephrine, Epi, and norepinephrine, NE) [29,30] which further increase glycogenolysis and modulate cardiovascular and respiratory function [31]. Since GCs regulate protein metabolism promoting catabolism, sustained high levels of GCs may lead to muscle atrophy emanating from reduced protein synthesis and elevated protein catabolism. Stress therefore compromises growth performance in fish since excessive proteolysis brings about muscle atrophy.

Changes in gene expression in response to various stressors such as, alkalinity, salinity [32], hypoxia [33], cold, thermal stress and crowding [34,35,36] have previously been reported in fish. Stressed fish have high cortisol and circulating glucose levels in the blood [37]. Fish under stress exhibit an increase in blood cortisol levels as well as a dose-dependent drop in hepatic insulin-like growth factor 1 mRNA levels. High cortisol has also been shown to increase hepatic insulin-like growth factor 1 binding protein 1(igfbp) 1b1 and -1b2 [38,39]. Crowding in Nile tilapia has been shown to increase the expression of somatostatin-1 (sst1), V-fos FBJ murine osteosarcoma viral oncogene homolog Ab (fosab) and prolactin genes [21,40]. There is however considerable variation in how fish respond to a stressor because of genetic differences among different taxa and also within stocks and species [41]. Growth hormone (gh), insulin-like growth factors (igf 1 and 2) and somatolactin (smtla) influence growth performance in tilapia [42]. This study showed that expression of these genes was positively correlated to growth while expression of myostatin (mystn) was negatively correlated. Mystn inhibits myoblast differentiation via Smad 3 and negatively regulates myogenesis by controlling myoblast proliferation [43,44]. Low myostatin expression has been associated with the double-muscling phenotype, while overexpression of the same is linked to muscle wasting [45,46].

Myogenic differentiation has been shown to be associated with an increase in insulin-stimulated glucose transport. After ammonia and stocking density stress, whole transcriptome analysis of differentially expressed genes in cultured nile tilapia (*O. niloticus*) was carried out. The goal of this study was to find potential genes and biological pathways linked with the response to chronic stress in (*O. niloticus*), which would throw further light on the adaptive and genetic processes involved in the response to chronic stress.

## 2. Materials and Methods

### 2.1. Ethical Statement

The Masinde Muliro University of Science and Technology Institutional Ethics and Review Committee (MMUST-IERC) (REF: MMU/COR: 403012 Vol 5 (01)) approved the animal procedures and all tests were carried out in accordance with the regulations.

### 2.2. Experimental Fish

Hand-sexed male Nile tilapia (*O. niloticus*) mean weight 25 ± 1.25 g, total length 8 ± 0.35 cm, were obtained from a local hatchery in Kakamega County, Kenya, and acclimatized to the laboratory conditions for 2 weeks prior to the start of the experiment [39]. The fish were maintained on a 12 h light and 12 h dark photoperiod cycle and were fed with commercial feed twice a day. The feeds used were of 2 mm diameter pellet size containing 32% of gross protein and 3500 Kcal/kg of digestible energy. The fish were fed to satiation for 30 min each time. The use of only male fish was intended to reduce unwanted variations that may result from sex differences.

### 2.3. Experimental Design

#### Ammonia Stress Experiment

A total of 525 healthy tilapia juveniles (25 ± 1.5 g) were randomly divided into seven groups and stocked in 21 white circular 500 L polyethylene tanks in a static system aerated by blowers connected to air stones [47]. Three replicates of 25 fish per tank were used. The tanks were fitted with calibrated thermostats set at 26 °C for temperature control. The dissolved oxygen concentration and temperature were measured with a digital oximeter (Hydrolab MSIP-REM-HAH-QUANTA (USA)) [48]. The dissolved oxygen was always maintained at levels above 6 mg/L while the temperature was maintained at 26 ± 1 °C for the period of the experiment. The pH was measured using a digital pH meter and maintained at 6.8 ± 0.2. The treatments were distributed as follows:

The first (control) group were maintained in natural water without addition of ammonia throughout the growth period. The ammonia concentration in the spring water was 0.12 mg/L. Six groups of 25 fish per tank were maintained at different concentrations of unionized ammonia (0.4, 0.8, 1.2, 1.6 and 2.0 and 2.4 mg/L) throughout the growth period. The different ammonia concentrations were set up by dissolving the appropriate amount of the ammonium chloride in each tank. Unionized ammonia concentration was determined using the formula by Emerson et al. (1975) [49].

Each of the treatments above were replicated thrice with the treatments being randomly assigned within the blocks; the fish were then reared for 70 days and their growth monitored every fortnight. Growth was monitored by determining the total length, weight, condition factor, growth rate and specific growth rate (SGR). During the growth period, the fish were maintained in a 12 h light:12 h dark photoperiod and fed to satiation three times a day with commercial feed containing 32% gross protein and 3500 Kcal/kg of digestible energy. During feeding, the fish were observed for thirty minutes and any feed remaining uneaten after 30 min was siphoned out and discarded. 

### 2.4. Stocking Density Stress Experiment

A total of 1135 juvenile tilapias (*O. niloticus*) were randomly distributed in 18 white circular polyethylene tanks (500 L). Each tank was assigned a treatment as follows: the control group were stocked at 25 fish per tank; six groups of fish were maintained at different stocking densities (40, 55, 70, 85 and 100 fish per tank).

Each of the treatments above was replicated thrice. Each of the treatments were randomly assigned within the blocks. A 50% water exchange was carried out every day to allow for the siphoning of the leftover feed and the excrement in each tank. The fish were then reared for 70 days and their growth monitored every fortnight. Growth was monitored by determining the total length, weight, condition factor, growth rate and specific growth rate (SGR). During the growth period, the fish were fed to satiation twice per day on commercial feed containing 32% gross protein and 3500 Kcal/kg of digestible energy. Satiation was decided by observing the fish while feeding for 30 min and any feed remaining uneaten was siphoned out.

### 2.5. Sample Collection

Three fish from each treatment were randomly sampled. The selected fish were anaesthetized in 3-aminobenzoic acid ethyl ester methane-sulfonate (MS-222, Sigma, St Louis, MO, USA) for physiological parameter measurements and scale removal for cortisol level determinations. The fish were then sacrificed and 2 g of muscle in the dorsal caudal region was quickly removed and frozen at −20 °C.

### 2.6. Sample Processing

#### Physiological Parameters

The total length and final live weight were determined and used to calculate the condition factor, growth rate and specific growth rate of fish in every treatment [50].

### 2.7. Growth Performance

Specific growth rate of the fish was determined by the formula: SGR (%) = (Ln final weight of fish − Ln initial weight of fish)/duration of experiment) ∗ 100 [50].

Relative Condition Factor Kn = (W/aL^b^) where ‘W’ is the body weight, ‘a’ the exponent relating rate of weight change with length of fish and ‘L’ the total length in cm of the fish.

### 2.8. RNA Extraction

Muscle tissue sampling was carried out by making an incision on the left side in the region between the lateral line, the dorsal and caudal fins. The tissue was quickly chopped into small pieces approximately 0.5 cm wide and quickly ground in liquid nitrogen. The ground tissue (1 g) was put in Eppendorf tubes containing 1 mL of TRIzol^®^ and vortexed for 30 s followed by a 10 min incubation at room temperature [51]. Phase separation was achieved by addition of 200 µL of chloroform. The mixture was then chilled on ice for 3 min and centrifuged at 12,000× *g* at 4 °C for 15 mins. The top layer of aqueous solution was carefully pipetted out and transferred to a 1.5 mL tube containing 500 mL of isopropanol and incubated for 10 min at room temperature. This was then centrifuged at 12,000× *g* for 10 min at 4 °C. After carefully removing the supernatant, 1 mL of 75% ethanol was used to wash the pellet. The mixture was then centrifuged at 7500× *g* at 4 °C for 5 min and the ethanol decanted. The wash step was repeated once and the pellet air-dried for 15 min at room temperature. The pellet was then dissolved in 50 µL of RNase-free water (Thermo Fisher Scientific, Waltham, MA, USA). To remove any possible genomic DNA contamination, the sample was treated with DNase1 (Thermo Fisher Scientific, Waltham, MA, USA). The quantity of RNA was then determined by measuring the absorbance at 260 nm and the quality determined from the 260/280 nm and 260/230 nm absorbance ratios with NanoVue (GE Healthcare, Chicago, IL, USA). Only samples with both 260/280 nm and 260/230 nm ratios above 2, were selected for sequencing. The three samples (replicates) were pooled by mixing equal volumes of RNA before carrying out cDNA synthesis. cDNA synthesis was carried out using SuperScript IV Reserve Transcriptase Kit (Thermo Fisher Scientific, Waltham, MA, USA). Second strand synthesis was carried out using NEBNext^®^ Ultra™ II RNA Library Prep Kit for Illumina^®^ (New England Biolabs GmbH, Ipswich, MA, USA). This was then purified using AMPure XP beads (New England Biolabs GmbH, Ipswich, MA, USA). Terminal repair, polyadenylation, sequencing adapter ligation, size selection, and degradation of second-strand U-containing cDNA were carried out using the Thermolabile USER^®^ II Enzyme (New England Biolabs GmbH, Ipswich, MA, USA). The strand-specific cDNA library was generated after the final PCR enrichment and quality assessment. The mRNA library was sequenced with Illumina Miseq (Illumina, Inc. US Illumina, San Diego, CA, USA) at Novogene Company Limited (Cambridge, UK).

### 2.9. Reads Mapping to the Reference Genome

Reference genome and gene model annotation files were downloaded from genome website directly. Hisat2v2.0.5 was used to build the index of the reference genome [52]. Paired-end clean reads were then aligned to the reference genome using Hisat2 v2.0.5. Hisat2 was chosen as the mapping tool because, in comparison to other non-splice mapping methods, it can create a database of splice junctions based on the gene model annotation file.

### 2.10. Quantification of Gene Expression Level

The reads mapped to each gene were counted using FeatureCountsv1.5.0-p3 [53], and the FPKM of each gene was then determined based on the length of the gene and the reads mapped to it. FPKM, expected number of Fragments Per Kilo base of transcript sequence per Millions base pairs sequenced, considers the effect of sequencing depth and gene length for the reads count at the same time, and is currently the most commonly used method for estimating gene expression levels. Differential expression analysis (for DESeq2 with biological replicates). Differential expression analysis of two conditions/groups (two biological replicates per condition) was performed using the DESeq2R package (1.20.0) [54]. DESeq2 provide statistical routines for determining differential expression in digital gene expression data using a model based on the negative binomial distribution. The resulting *p*-values were adjusted using the Benjamini and Hochberg’s approach for controlling the false discovery rate. Genes with an adjusted *p*-value ≤ 0.05 found by DESeq2 were assigned as differentially expressed (for edge R without biological replicates). Prior to differential gene expression analysis, for each sequenced library, the read counts were adjusted by edge R program package through one scaling normalized factor. Differential expression analysis of two conditions was performed using the edge R package (3.22.5). The *p*-values were adjusted using the Benjamini and Hochberg method. Corrected *p*-value of 0.05 and absolute fold change of 2 were set as the threshold for significantly differential expression. Gene Ontology (GO) enrichment analysis of differentially expressed genes was implemented by the Cluster Profiler R package [55], in which gene length bias was corrected. GO terms with corrected *p*-value less than 0.05 were considered significantly enriched by differential expressed genes. KEGG is a database resource for understanding high-level functions and utilities of the biological system, such as the cell, the organism and the ecosystem, from molecular-level information, especially large-scale molecular data sets generated by genome sequencing and other high-through put experimental technologies (http://www.genome.jp/kegg/ accessed on 20 August 2021). Cluster Profiler R package was used to test the statistical enrichment of differential expression genes in KEGG pathways. The Reactome database brings together the various reactions and biological pathways of human model species. Reactome pathways with corrected *p*-value less than 0.05 were considered significantly enriched by differential expressed genes. The DO (Disease Ontology) database describes the function of human genes and diseases. DO pathways with corrected *p*-value less than 0.05 were considered significantly enriched by differential expressed genes. The DisGeNET database integrates human disease-related genes. DisGeNET pathways with corrected *p*-value less than 0.05 were considered significantly enriched by differential expressed genes. Cluster profiler software was used to test the statistical enrichment of differentially expressed genes in the Reactome pathway, the DO pathway, and the DisGeNET pathway.

### 2.11. Real Time qPCR

To verify the reliability of the RNA-Seq results obtained, RT q-PCR was performed on four selected genes. From the two treatments, one DEG from each of the up- and down-regulated DEGs was randomly chosen. Four DEGs (DUSP-1 and Mych (down-regulated), Pfkma, and Acetyl CoA Thioesterase 4 (up-regulated)) were the randomly selected DEGs for RT-qPCR. RNA was extracted using TRIzol reagent (Invitrogen, Washington, Waltham, MA, USA). This was used to prepare a cDNA library using RevertAid First strand cDNA Synthesis Kit (Thermo Fisher Scientific, Waltham, MA, USA) following the manufacturer’s instructions. The reaction mixture consisted of 10 µg template, 2µL dNTPs (Thermo Fisher Scientific, Waltham, MA, USA), 2 µL of 10×Dream Taq buffer (Thermo Fisher Scientific, Waltham, MA, USA), 0.25 µL of 5 U/µL Dream Taq DNA, 20×Evagreen^®^Dye (BIOZOL Diagnostica Vertrieb GmbH, Eching, Germany) [56], 1 µL of 10 pM forward and reverse primers and topped up to 20 µL with water. The primers used (Table 1) were designed using primer3plus [57]. Real time qPCR was performed on CFX Connect Real-Time PCR optical detection System (Bio-Rad, Hercules, CA, USA). The qPCR amplification products of the four genes were normalized with those of glyceraldehyde-3-phoshate dehydrogenase gene. One-way ANOVA was used to determine the significance difference between the levels of expression of the ammonia, stocking density treatments and the untreated controls.

## 3. Results

### 3.1. Growth Performance

To determine the growth performance of Nile tilapia subjected to stress, length and weight measurements were carried out. These were used to calculate the specific condition factor and specific growth rates (SGR). Higher ammonia concentrations resulted in significantly lower SGR values than the controls (Table 2). The stocking density treatment also replicated a similar observation (Table 3). The growth rates were positively correlated with the relative condition factor (K_n_) in both cases. The K_n_ decreased with increase in ammonia concentrations as well as increase in stocking densities.

To determine the impact of stress on the weight gain of Nile tilapia under chronic stress, a plot of weight against time was drawn (Figure 1A,B). In the ammonia treatment it is evident that the higher the ammonia concentration, the lower the weight gain (Figure 1A). Similarly, higher stocking densities resulted in slower weight gain rates (Figure 1B). At the onset of the experiment, the growth was almost the same for all the treatments. For ammonia treatment, the difference in weight gain began to be evident from the fourth week while the fish under stocking density treatment showed the difference from the sixth week. During the period of experimentation, eight fish died. The fish that died were distributed in the various tanks as follows; two fish in the tanks with 2.4 mg/L ammonia, two from the 1.2 mg/L tank and one from the 1.6 mg/L tank. From the stocking density treatment, two fish died in the 100 fish per tank treatment and one died in the 55 fish per tank experiment. The reported mortality, however, did not significantly affect the outcome of the experiment.

Raw data were filtered, generating 47,203,186 to 53,152,344 clean reads in the ammonia treatment and 45,860,616 to 54,721,982 in the stocking density treatment. All the samples had Q20 and Q30 values greater than 97.01 and 92.01, respectively, and a GC content between of 50.01 and 50.95. When compared to the reference genome i.e., from *O. niloticus*, the ratio of the mapped reads was approximately 92%. Results from the principal component analysis (PCA) shows that the samples clustered into the controls consistent with the sample grouping. These results are an indication that the data from this analysis were of high quality and suitable for DEGs and other downstream analysis.

To identify the regulated genes, volcano plots were used. A combined volcano plot for the two treatments revealed myosin light chain 13, myosin 7 and acyl-coenzyme thioesterase 4 as the most statistically significantly up-regulated genes, while potassium voltage-gated channel subfamily A regulatory β subunit 1b was the most enriched gene (Figure 2A).

A total of 20,483 genes were identified from the ammonia treatment samples (Figure 2B). On the other hand, 19,781 genes were identified in the stocking density treatment (Figure 2C).

Comparative transcriptome analysis revealed the proportion of regulated genes. Ammonia treatment had a total of 209 differentially expressed genes (DEGs), with 156 being up-regulated and 53 down-regulated. On the other hand, stocking density treatment had 252 DEGs, of which 175 were up-regulated and 77 down-regulated (Figure 3).

Among the up-regulated genes, 132 genes were unique to ammonia treatment, 151 genes were unique to stocking density, and 24 of them were common to both treatments (Figure 4A). On the other hand, considering the down-regulated genes, the ammonia treatment had 36 unique genes, stocking density treatment had 60, and 17 down-regulated genes were common to both treatments (Figure 4B).

Gene ontology was performed to classify the DEGs according to their functions. The findings of the GO enrichment study of the DEGs were divided into three GO categories based on Biological Processes (BP), Cellular Components (CC) and Molecular Function (MF). The significantly up-regulated DEGs were mainly enriched in “Oxidation reduction processes” of BP, “cytoskeletal part” and “cytoskeleton” of CC and “motor activity” of MF (Figure 5A), while the significantly down-regulated DEGs were mainly enriched in “Nucleosome assembly” of BP, “Nucleosome” of CC and “protein kinase activity” of MF (Figure 5B) in ammonia treatment. The up-regulated DEGs were mainly enriched in “regulation of metabolic process” of BP, “MHC protein complex” of CC and “Sequence specific DNA binding” of MF (Figure 5C), while the down-regulated DEGS were “nucleosome assembly” of BP, “nucleosome” of CC and “Oxygen binding” of MF (Figure 5D) in the stocking density treatment.

The most enriched pathways were mainly “Oxidation reduction processes” and “carboxylic acid” of BP, “cytoskeletal part I” and “cytoskeleton” of CC and “motor activity” and “pyrophosphate activity” of MF in ammonia treatment (Figure 5A), while “immune response” and “immune system process” of BP, “MHC class II” and “MHC protein complex” of CC and “Sequence specific DNA binding” and “Transcriptome regulation activity” of the MF were the most enriched pathways in the stocking density treatment (Figure 5B). To understand the function of the significant DEGs in the signaling pathways, we annotated the significantly up- and down-regulated DEGs in the ammonia and stocking density treatments in the KEGG database.

To better understand the DEGs in the individuals assigned to the two treatments, the ratio between the level of enrichment between the up-regulated and the down-regulated genes in the three categories was assessed (Figure 6). In the ammonia treatment, the immune response of MF, MHC protein complex of CC and sequence-specific DNA binding of BP showed the highest difference (Figure 6A). In the stocking density treatment, oxidation reduction process of MF, cytoskeleton of CC and motor activity of BP showed the highest difference (Figure 6B).

To identify the significantly enriched pathways, the identified DEGs were then subjected to KEGG analysis (Figure 7). The most significantly enriched pathways in ammonia treatment were fatty acid elongation, glycolysis/gluconeogenesis, fatty acid degradation, adrenergic signaling in cardiomyocytes and cardiac muscle contraction (Figure 7A). On the other hand, stocking density treatment showed the intestinal immune network for IGA production, cardiac muscle contraction, adrenergic signaling in cardiomyocytes, cell adhesion molecules and phagosome as the most significantly enriched pathways (Figure 7B).

KEGG pathways were used to show the position of the DEGs in the metabolic pathways to understand their function in phenotype related to growth performance. The significantly enriched DEGs were mainly involved in muscle activity, energy mobilization and immune-related functions (Figure 8, Figure 9 and Figure 10).

### 3.2. Cardiac Muscle Signaling Pathway

The cardiac muscle contraction pathway is responsible for the contractility of the heart and consequent pumping motion. This pathway was significantly up-regulated in ammonia treatment. Ammonia-induced stress up-regulated 16 DEGs in this pathway while down-regulating one DEG in the same. In the stocking density treatment, the same pathway was up-regulated, with 16 DEGs being up- and 1 DEG being down-regulated (Figure 8).

### 3.3. Adrenergic Signaling in Cardiomyocytes Pathway

Among the downstream effects of the activation of the adrenergic signaling pathway is the stimulation of apoptosis, and enhanced contractility of cardiac muscles and its speed of contraction. This pathway was significantly up-regulated in ammonia treatment. Ammonia-induced stress up-regulated 16 DEGs and down-regulated 2 DEGs in this pathway (Figure 9).

### 3.4. Fatty Acid Degradation

β-oxidation is an important metabolic energy source during interprandial periods and at times of high energy demand, such as exercise [57]. The fatty acid degradation pathway enables access to the primary energy source for animals, adenosine triphosphate (ATP), by breaking down fatty acids. This pathway was significantly up-regulated in ammonia treatment. Ammonia-induced stress up-regulated four DEGs in this pathway (Figure 10).

### 3.5. Validation of RNA-Seq Results

To verify the DEGs identified, by RNA-seq, four DEGs were subjected to RT-qPCR of which two were down-regulated (Dual Specific Phosphatase 1 (DUSP 1) and Myelocytomatosis Oncogene Homolog (MYCH)) and two were up-regulated (Phosphofructokinase Muscle A (PFKMA) and Acyl-CoA Thioesterase 4 (AcoT4)) (Figure 11). The gene glyceraldehyde phosphate dehydrogenase (GAPDH) was used as the house keeping gene. It is clear that changes in RT-qPCR expression alter in the same way. The RT-qPCR data and the Illumina sequencing findings were in agreement, demonstrating the validity and accuracy of the transcriptome analysis. Through this analysis, the results from RNA-seq guarantee that DEGs that were discovered under stress and that additional research into these or other DEGs from the transcriptome data is viable and sustainable.

## 4. Discussion

Growth performance is an important aspect governing the choice of fish for any aquaculture establishment. Fish genetics, feed quality and culture environment are key determinants of growth performance. While genetics and feeds have received considerable attention, little has been investigated with regards to the influence of fish culture water to fish growth performance. Fish raised in captivity develop in an environment that frequently becomes stressful. High stocking densities and ammonia are the two main chronic stressors present in many aquaculture facilities, informing their choice in this study. Chronic stress resulted in depressed growth in juvenile *O. niloticus.* Fish raised in environments with greater ammonia concentrations exhibited slower weight gain (Figure 1A) compared to the controls. Similarly, higher stocking density resulted in slower weight gain (Figure 1B). On the other hand, fish raised in higher ammonia concentrations had significantly lower SGR values than the controls (Table 2). A similar observation was replicated in the stocking density treatment (Table 3). The growth rates were positively correlated with the relative condition factor (K_n_) in both cases. This is consistent with the assertion of [58] that higher K_n_ values are indicative of fish wellness. High SGR and K_n_ values observed in the control fish indicate that higher growth is associated with fish wellness. Typically, a fish’s stress reaction is characterized by an increase in muscular activity [59]. The cardiac muscle contraction, the calcium signaling pathway, and the adrenergic signaling in cardiomyocytes are important indicators of muscle function. Results of this study indicate that these processes are enhanced in chronic stress scenarios. Muscle functioning requires the expenditure of energy in the form of ATP. Energy mobilization is significantly influenced by metabolic processes such as fatty acid breakdown, elongation, production of unsaturated fatty acids, and glycolysis/gluconeogenesis. The findings of this study document that these pathways are also enhanced in chronic stress. The reduction in growth performance of the chronically stressed fish compared to their non-stressed counterparts (controls) may be due to the divergence of energy from muscle development to muscular activities. Energy is redirected to organs, such as the brain and the skeletal muscles that react to stress by becoming more active. Stressed fish have higher ventilation rates compared to unstressed controls and these rates have been used as a tool for assessing both alertness and stress levels in fish [60].

Results from the current study indicate that chronic stress up-regulated the cardiac muscle signaling pathway and the adrenergic signaling in cardiomyocytes pathway. In these two pathways, 15 common DEGs were significantly up-regulated. The cardiac muscle contraction pathway has an additional DEG (Cytochrome C oxidase subunit 6B1, which is key in electron transport chain) while the adrenergic signaling in cardiomyocytes has MAPK12a being up-regulated. These up-regulated common DEGs are involved in facilitating muscle contraction. Some of the DEGS that were up-regulated in this pathway include ATPase sarcoplasmic/endoplasmic reticulum Ca^2+^ transporting 2a (ATP2a2a) and ATPase sarcoplasmic/endoplasmic reticulum Ca^2+^ transporting 1 (ATP2a1), which are involved in the transport of calcium into the lumen of sarcoplasmic reticulum [61]. Cardiac myosin light chain-1 (cmlc1), which enables calcium binding, is also up-regulated. Calcium binding onto troponin brings about conformational changes on tropomyosin, allowing the binding of myosin heads to actin to initiate muscle contraction [59]. In the current study, tropomyosin 2 (β) and tropomyosin α-3 chain are up-regulated. These two proteins are involved in stabilization of the binding of actin and myosin, regulating the tensing of the muscle fiber. Cytochrome C oxidase subunit 6B1 and Na (+)/H (+) exchanger β are also significantly enriched in this study. Cytochrome C oxidase subunit 6B1 is key in the electron transport chain, catalyzing the transfer of electrons to molecular O_2_ and driving oxidative phosphorylation. The Na (+)/H (+) exchanger β, on the other hand, maintains intracellular pH [62] by exporting protons and importing Na+. The cardiac muscle signaling pathway and the adrenergic signaling pathway complement each other in promoting muscular contraction [63]. The up-regulation of both cardiac muscle contraction and adrenergic signaling in cardiomyocytes is beneficial to the blood pumping process. Ref. [58] demonstrated that adrenergic activation enhances heart performance in acute thermal stress in various teleost.

Results of the current study indicate that these two pathways are up-regulated in chronic stress. Chronic stress has been shown to raise heart rate and blood pressure, exerting more strain on the heart’s ability to pump blood to the body’s various organs [64,65]. The heart has a high energy requirement; in order to maintain its contractile function, it must continually produce large amounts of ATP. This ATP may be produced using a variety of substrates, such as fatty acids, carbohydrates, proteins, and ketones [65,66]. This process demands high amounts of oxygen, resulting in the heart consuming more oxygen per unit weight than any other organ in the body [66]. According to [60], stressed fish show higher ventilation rates compared to unstressed controls.

Mitogen-activated protein kinase 12a (mapk12a) is thought to be involved in the calcium-independent control of smooth muscle contraction in differentiated smooth muscle. On the other hand, stocking density also up-regulated cyclic AMP (cAMP) responsive element binding protein 1, which functions as an activator of the phosphorylating enzyme, protein kinase A (PKA) [67] and β 2 adrenergic receptor, which mediates smooth muscle relaxation [68]. PKA is an important energy-regulating enzyme which is responsive to the cellular levels of cAMP [69]. The findings of this study have also demonstrated that chronic stress down-regulates actin, α cardiac, the primary protein of the thin filament in cardiac sarcomeres, actin α I skeletal muscle a (acta1a) an ATP binding protein in skeletal muscle and cAMP responsive element modulator a (crema). Acta1a activates numerous other “myogenic genes”, which are crucial for the development of muscles [70]. Crema participates in a number of intracellular activities, including neurogenesis, synaptic potentiation, differentiation, proliferation, and neuroplasticity [71,72]. Taken together, the adrenergic signaling pathway promotes muscular contraction, which is an energy-expensive process, and down-regulates the formation of muscle fiber, compromising muscle growth. This is consistent with the findings of the growth performance analysis for the current study, which demonstrated that the experimental fish in the ammonia and the stocking density treatments both developed more slowly than the controls.

Alcohol dehydrogenase 1 (ADH1), which facilitates the inter conversion of alcohols and aldehydes or ketones with the reduction of nicotinamide adenine dinucleotide (NAD+) to NADH [72], Carnitine palmitoyl transferase 1Ab (cpt1ab), which catalyzes the reversible exchange of acyl groups (which is derived from fatty acids) between coenzyme A, and the acyl-CoA synthetase long chain family member 1a, which transforms free long chain fatty acids into fatty acyl-CoA esters and is involved in both lipid biosynthesis and fatty acid degradation [73], were the three DEGs in the fatty acid degradation pathway that were up-regulated by increased ammonia levels. Chronic stress also up-regulated the biosynthesis of unsaturated fatty acids and fatty acid elongation pathways. The DEGS that were up-regulated in these pathways are: acyl-coenzyme A thioesterase 4 (ACOT 4), acyl-coenzyme A thioesterase 1 (ACOT 1), ELOVL family member 6 (Elovl6l) and elongation of long chain fatty acids were the significant DEGs that were up-regulated in this pathway and are involved in the regulation of intracellular levels of CoA esters. Carnitine palmitoyltransferase 1Ab and alcohol dehydrogenase 1 are also significantly up-regulated. ACOT4 plays a crucial role in both fatty acid and peroxisomal lipid metabolism. ACOT 4 is a paralog of ACOT1. Biosynthesis of unsaturated fatty acids, fatty acid elongation and fatty acid degradation DEGs are significantly up-regulated in stress. The levels of acetyl CoA determine which processes, biosynthetic or degradation, will occur. Acetyl-CoA carboxylase, the only controlled and the primary enzyme in fatty acid synthesis, is phosphorylated/dephosphorylated by both AMP-activated protein kinase (AMPK) and Protein Kinase A. De-phosphorylation of Acetyl-CoA carboxylase activates it and promotes fatty acid biosynthesis while its phosphorylation promotes fatty acid degradation. AMP Kinase is an energy status sensor that controls cellular and overall energy balance in the body [74]. AMPK maintains the energy balance by regulating metabolic processes such as mitochondrial synthesis, autophagy, fatty acid oxidation, glucose and lipid transport, and oxidative metabolism while decreasing ATP-consuming processes such as glucose, proteins, fatty acids and cholesterol synthesis [75]. Results from the current study indicate that under chronic stress, AMPK is up-regulated, thus promoting fatty acid degradation. Chronic stress was also seen to down-regulate γ enolase, a key enzyme in glycolysis and gluconeogenesis. γ enolase is particularly active in cells undergoing increased aerobic glycolysis [76]. In this study, γ enolase was significantly down-regulated, pointing to the fact that glycolysis was down-regulated, and hence fatty acid degradation may be the principal source of metabolic energy during stress. Results of the current study also indicated a significant increase in blood glucose level in stressed fish compared to the controls. The decreased utilization of glucose may therefore be associated with this accumulation of glucose in the blood. Several other studies [77,78,79] have also associated chronic stress with hyperglycaemia in fish. Activation of AMPK has also been shown to influence the functioning of Na+/H+ exchanger, promoting the export of H+ out of the cell in exchange of Na^+^ [80,81]. In the current study, both AMPK and Na+/H+ exchanger were significantly up-regulated. This predisposes the cell to degradative pathways rather than anabolic pathways and hence growth is compromised. Growth performance analysis conducted for the current study revealed that the experimental fish in the ammonia and stocking density treatments both grew more slowly than the unstressed controls.

The significant up-regulated DEGs annotated to the apelin signaling pathway include the myosin light chain kinase family, member 4a (mylk4a), myosin light chain 2b regulatory cardiac slow (myl2b), and the early growth response 1 (egr1) a regulator of cell proliferation and apoptosis. Myosin regulatory light chains are phosphorylated by the enzymes mylk4a and myl2b to help myosin connect with actin filaments and induce contractile activity [82]. The up-regulation of mylk4a and myl2b is evidence of heightened muscle activity which in turn leads to higher energy consumption. Myosin, light chain 13(myl13), cardiac myosin light chain-1(cmlc1) and ryanodine receptor 1 were up-regulated in this pathway in ammonia treatment and are actively involved in muscle activity. Na (+)/H (+) exchanger β helps to maintain the correct pH for muscular activity [81,83,84]. On the other hand, cadherin-1, a gene involved in cell–cell adhesion and of fundamental importance in the development and maintenance of tissues, was down-regulated in the stocking density treatment. Taken together, higher consumption of energy and reduced cell–cell adhesion results in depressed growth performance in stressed fish. The apelin signaling pathway was up-regulated in chronic stress. Numerous physiological processes and pathological conditions, such as cardiovascular disease, angiogenesis, energy metabolism, and fluid balance, are influenced by the apelin/APJ system [85]. Due to its involvement in muscle movement, cooperatively with cardiac signaling pathway and adrenergic signaling pathway, we can infer that chronic stress increases muscular activity and thus energy consumption at the expense of growth. This is in concurrence with the growth performance results in the current study, which show that the experimental fish in both ammonia and stocking density treatments had a lower growth rate as compared to the controls. A previous study [86] demonstrated an increase in apelin with fasting (stress) in goldfish (*Carassius auratus*). Apelin signaling has also been shown to affect the intake of food and water, as well as the central and peripheral control of the cardiovascular system [83].

Calcium signaling pathway was also significantly up-regulated in both ammonia and stocking density treatment. The function of cardiac myocytes is significantly regulated by Ca^2+^. Calcium is primarily the connection between electrical impulses that are sent throughout the heart and myocyte contraction to move blood [84]. A number of contractile proteins are activated and deactivated by Ca^2+^ in the cytosol during contraction and relaxation [82]. DEGs such as 5-hydroxytryptamine receptor 7, myosin light chain kinase family member 4a (mylk4a), fibroblast growth factor receptor 4 (fgfr4), adenosine A2a receptor a (adora2aa), immunoglobulin mu heavy chain-like and β-2 adrenergic receptor were up-regulated in the calcium signaling pathway in the stocking density treatment and are involved in muscle activity and signal transduction. Ammonia treatment up-regulated seven DEGs, including solute carrier family 25 member 5 (slc25a5), troponin c slow skeletal and cardiac muscles, ATPase Sarcoplsmic/Endoplasmic reticulum Ca^2+^ transporting 1 (ATP2a1), Ryanodine Receptor 1, troponin c type 1a (slow) (tnnc1a) and 5-hydroxytryptamine receptor 7. These DEGs are important components of muscle contraction and relaxation cascade. Altogether, the pathways discussed above involved in muscle activity and energy mobilization. The effect of increased muscle activity and thus energy consumption accompanying chronic stress attenuates growth, as evidenced in this study. A previous study by [87] reported the up-regulation of Ca^2+^ signaling pathway and troponins and glycolytic genes in superior performing hybrid grouper. These were associated with enhanced muscle activity and growth. In the current study, enhanced muscle activity in stress correlates to depressed growth. Several studies on growth performance have focused on the insulin-like growth factors, their receptors, growth hormones and their receptors, myostatin and somatolactin [42,88,89,90]. Most of these studies involved acute stress. In the current study, these genes were not significantly up-regulated. It remains to be found out whether they could have fallen back to basal levels with time.

Unlike chronic stress, acute stress has received tremendous research interest due to its calamitous effects. Even though chronic stress may not result in fish kills, the economic losses arising from lost productivity and mitigation measures are enormous. Furthermore, the deterioration of fish flesh quality after slaughter in terms of color, flavor, firmness, weight loss/water holding capacity and perishability have been documented [91]. Many studies on chronic stress have focused on stocking density, temperature and anoxia. Few studies have attempted to explain the molecular basis of depressed growth in fish under chronic ammonia exposure. This is the first report ever attempting to decipher how sub lethal concentrations of ammonia affect *O. niloticus* growth. Results of the current study add to the number of molecular mechanisms explaining growth depression associated with chronic stress.

Ammonia treatment up-regulates 16 DEGs in the cardiac muscle signaling pathway, with three of these DEGs being also up-regulated in the stocking density treatment. Three DEGs in this pathway were significantly down-regulated in stocking density treatment while only one DEG was down-regulated in ammonia treatment. In the adrenergic signaling pathway, ammonia treatment up-regulated 16 DEGs and down-regulated 2 DEGs while the stocking density treatment only up-regulated 5 DEGs and down-regulated 3 DEGs in the same pathway. Ammonia treatment up-regulates muscle activity related genes as compared to stocking density treatment.

Results from this study indicate that stocking density up-regulated the intestinal immune pathway. Mamu histocompatibility antigen, Histocompatibility class II antigen and immunoglobulin Mu heavy chain were significantly up-regulated in this pathway. On the other hand, only immunoglobulin mu heavy chain was up-regulated in ammonia treatment. Eight DEGs were significantly up-regulated in the phagosome pathway and one was significantly down-regulated in the stocking density treatment, while four DEGs were significantly up-regulated and one down-regulated in ammonia treatment. Class I histocompatibility antigen and immunoglobulin MU heavy-chain-like genes were significantly up-regulated in both ammonia and stocking density treatments.

Interleukin 8 (IL-8) was also significantly down-regulated in this study. Similar findings were obtained when *O. mykiss* was subjected to increased stocking densities [89]. In this study, ammonia down-regulated IL-8, though not significantly. A study by [92] showed that acute ammonia stress up-regulates IL-8 in Pacific white shrimp (*Litopenaeus vannamei*).

Stocking density down-regulated Super oxide dismutase 1 (SOD1) and catalase (CAT), while it up-regulated SOD2 and SOD3, but not significantly. In *Labeo lohita*, SOD and CAT were up-regulated when subjected to increased stocking density. In Chinese sturgeon, SOD and IL-8 were significantly down-regulated by increasing stocking density. In ammonia treatment, results from this study indicate that SOD 1, 2 and 3 were down-regulated, though not significantly. Catalase was up-regulated in this treatment, but not significantly. In Pacific white shrimp, ammonia has been shown to down-regulate SOD and catalase [93]. In puffer fish (*Takifugu obscurus*), ammonia up-regulated catalase and SOD [94]. These observations amplify the findings of [95], in that stress reaction varies according to type and intensity of stressor as well as the species involved and the stage of development.

Taken together, stocking density up-regulates immune-related genes. At low stocking densities, tilapia immunity was impaired according to a study by [96]. The study also showed that increasing stocking density enhances the expression of innate immune genes in Atlantic salmon.

## 5. Conclusions

The findings of this study provide plausible grounds to show that chronic ammonia stress increases muscle activity while chronic stocking density stress activates immune-related pathways. These reactions use significant amounts of energy. The depressed growth witnessed in the stressed fish in this study is a reflection of the diversion of energy from growth to fighting chronic stress. This study sheds light on the different mechanisms that fish use to survive chronic stress depending on the uniqueness of the stressor. Taken together, cultured Nile tilapia suffer depressed growth when reared in sub-optimal water environments, leading to potentially huge economic losses for fish farmers. This study affirms the calamitous consequences of high ammonia concentrations on growth performance and recommends its concentration be maintained at levels below 1.2 mg/L in the fish grow out ponds.

## Figures and Tables

**Figure 1 genes-14-00795-f001:**
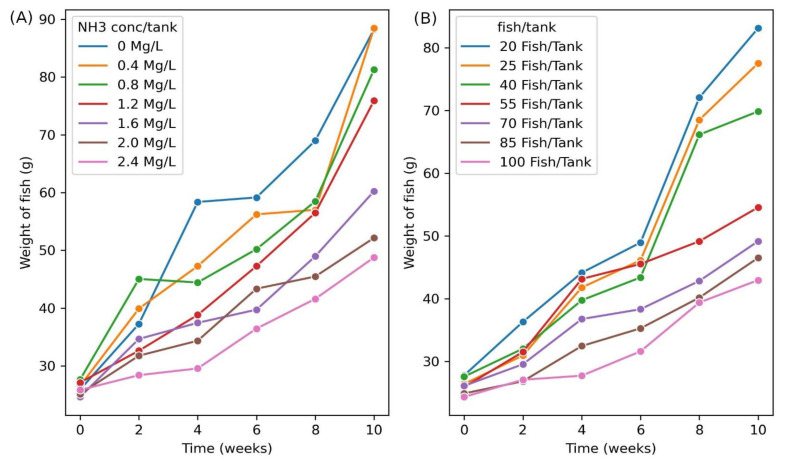
Fortnight weight gain for Nile tilapia subjected to chronic stress. (**A**) Weight gain for fish in ammonia treatment. Each line represents the average variation of weight with time at the different ammonia concentrations. (**B**) Weight gain for fish in stocking density treatment. Each line represents the average variation of weight with time at the different stocking densities.

**Figure 2 genes-14-00795-f002:**
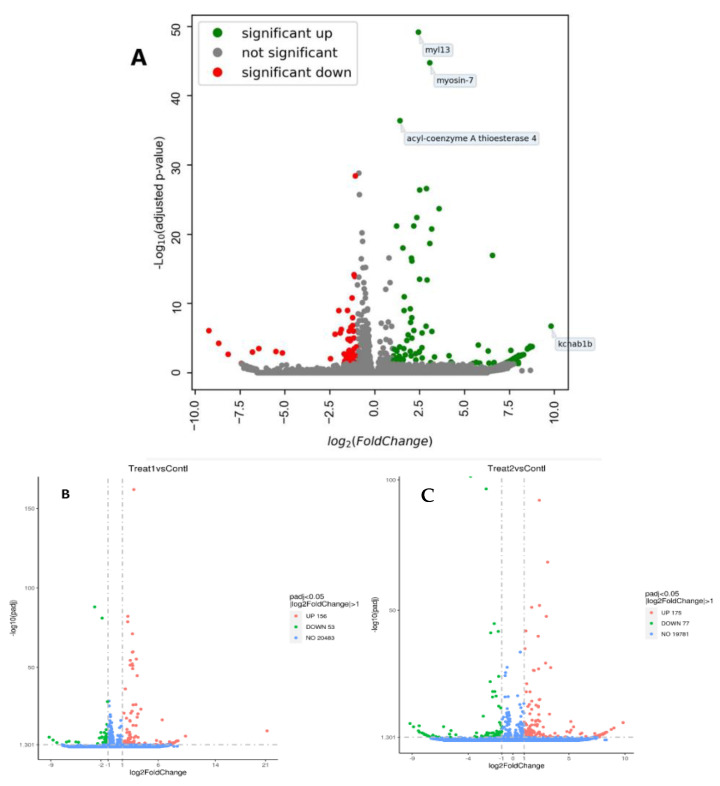
Volcano plot showing the distribution of DEGs in *O. niloticus* subjected to chronic stress. (**A**) Combined volcano plot for ammonia and stocking density treatments. The red dots represent the down-regulated genes, the green dots represent the up-regulated genes and the grey dots represent those that are not significantly regulated in the two treatments. (**B**) Volcano plot for ammonia treatment. The green dots represent the down-regulated genes, the red dots depict the up-regulated genes and the blue ones represent the genes that are not significantly up- or down-regulated under high ammonia concentrations. (**C**) Volcano plot for the stocking density treatment. The green dots represent the down-regulated genes, the red dots depict the up-regulated genes and the blue ones represent the genes that are not significantly up- or down-regulated under different stocking densities.

**Figure 3 genes-14-00795-f003:**
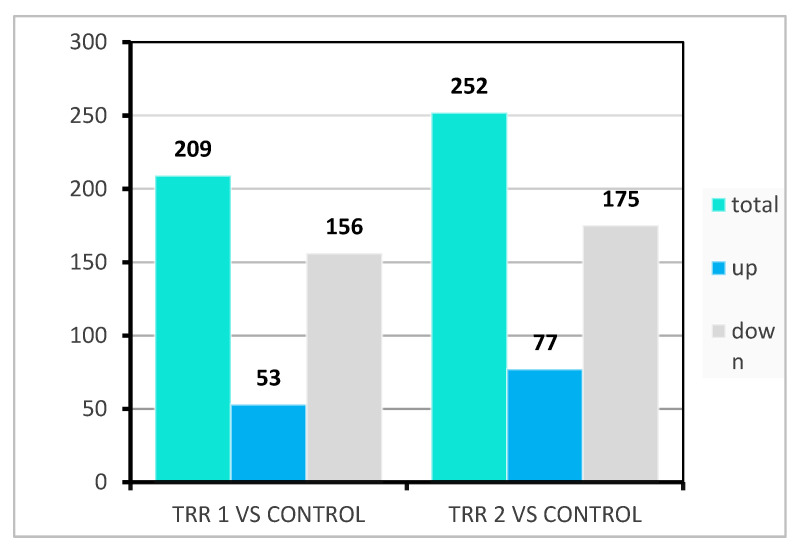
Differentially expressed genes in ammonia (treatment 1) and stocking density (treatment 2) compared to the controls. The threshold for a gene to be considered differentially expressed was DESeq2 padj ≤ 0.05 and |log2FoldChange| ≥ 1.0.

**Figure 4 genes-14-00795-f004:**
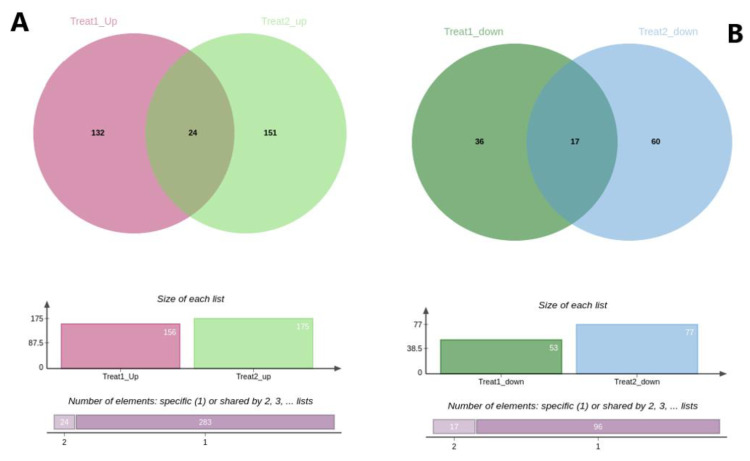
Venn diagram displaying the number of genes that were up- and down-regulated in the ammonia and stocking density treatments. (**A**) Up-regulated genes in ammonia and stocking density treatment. Pink color represents ammonia treatment, green represents stocking density while the intersection represents the common elements. (**B**) Down-regulated genes in ammonia and stocking density treatment. Green color represents ammonia treatment, blue color represents stocking density treatment and the intersection represents the common elements.

**Figure 5 genes-14-00795-f005:**
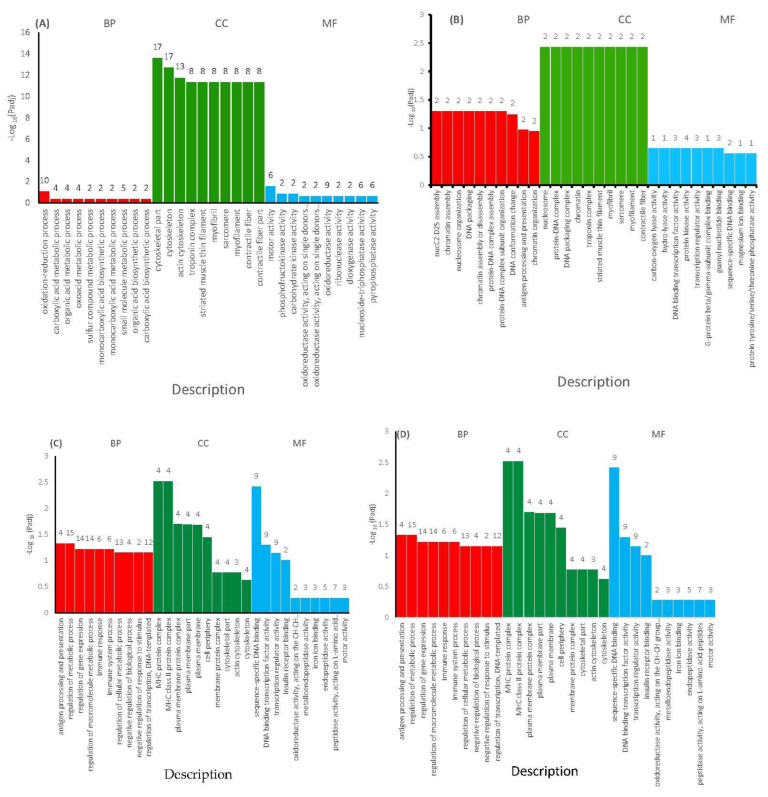
Gene ontology functional enrichment analysis for the differentially expressed genes under chronic stress. (**A**) The 10 most up-regulated DEGs in the Biological Processes, Cellular Components and Molecular Function categories in ammonia treatment. (**B**) The 10 most down-regulated DEGs in the Biological Processes, Cellular Components and Molecular Function categories in ammonia treatment. (**C**) The 10 most up-regulated DEGs in the Biological Processes, Cellular Components and Molecular Function categories in stocking density treatment. (**D**) The 10 most down-regulated DEGs in the Biological Processes, Cellular Components and Molecular Function categories in stocking density treatment.

**Figure 6 genes-14-00795-f006:**
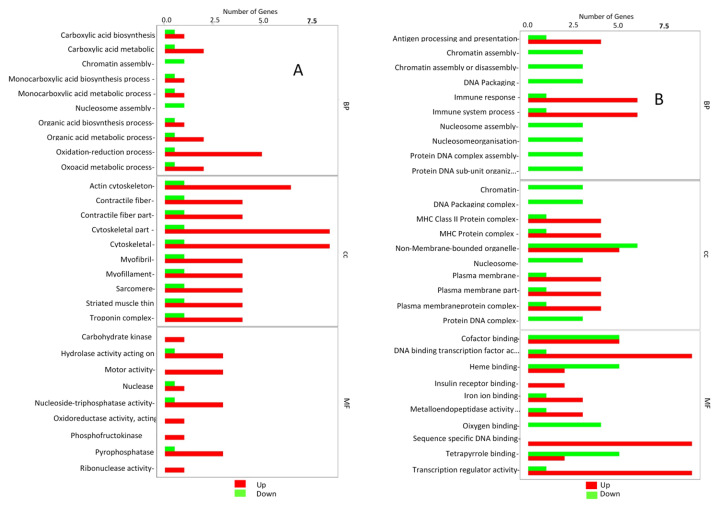
Gene Ontology analysis bar plots for the most significantly enriched pathways under chronic stress. (**A**) Ammonia treatment showing the 10 most significantly enriched pathways in the Cellular component, Biological Process and molecular Function categories. The green bars represent the down-regulated genes while the red bars represent the up-regulated genes. (**B**) Stocking density treatment showing the 10 most significantly enriched pathways in the Cellular component, Biological Process and molecular Function categories. The green bars represent the down-regulated genes while the red bars represent the up-regulated genes. The proportion of the red (up-regulated) to the Green (down-regulated) depicts the level of pathway enrichment.

**Figure 7 genes-14-00795-f007:**
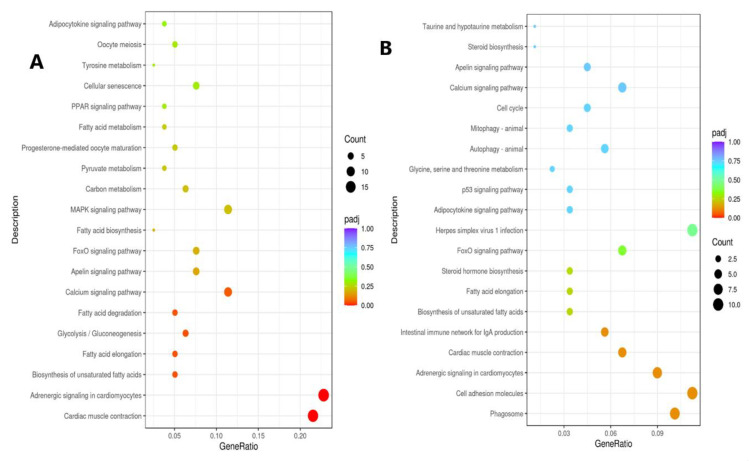
Dot plots representing the results of KEGG analysis showing the most significantly up- and down-regulated pathways under chronic stress. (**A**) KEGG analysis for the ammonia experiment showing the most significantly up- and down-regulated pathways. (**B**) KEGG analysis for the crowding experiment showing the most significantly up- and down-regulated pathways.

**Figure 8 genes-14-00795-f008:**
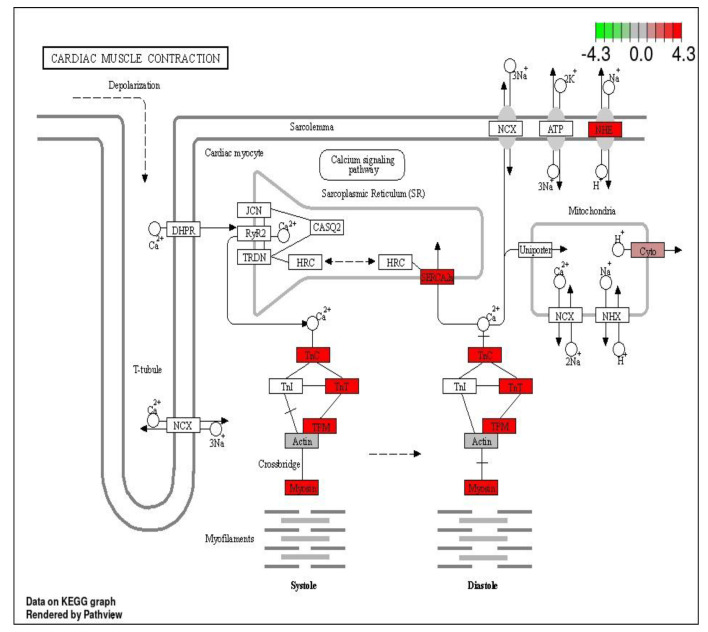
KEGG plot showing the cardiac muscle contraction pathway showing the pathway terms that are enriched in ammonia treatment. The red border shows the enriched terms.

**Figure 9 genes-14-00795-f009:**
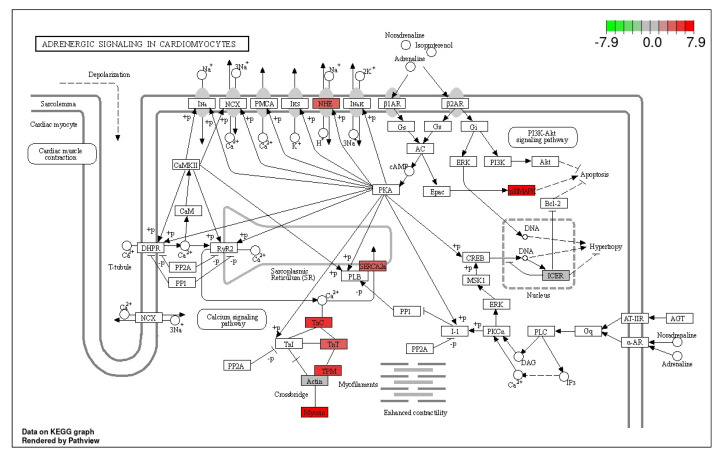
Adrenergic signaling pathway showing the pathway terms that are enriched in ammonia treatment. The color shades refer to the extent of up-regulation.

**Figure 10 genes-14-00795-f010:**
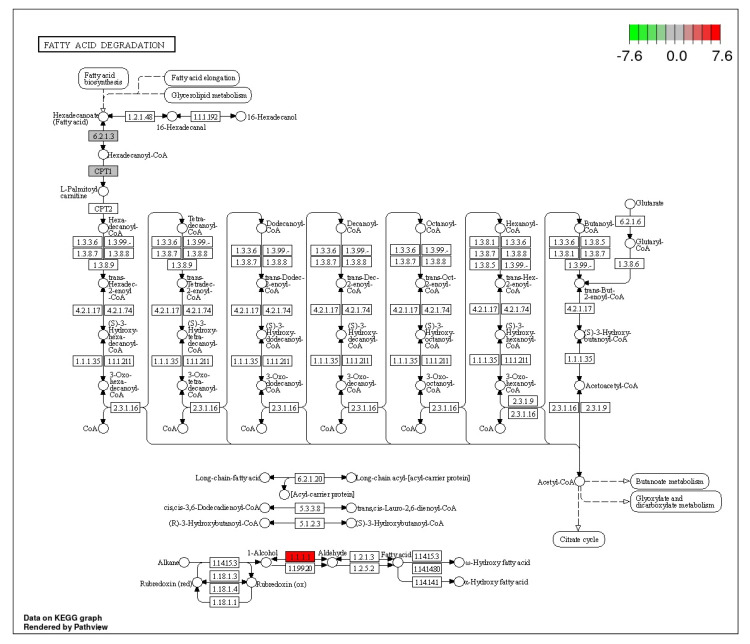
Fatty acid degradation pathway showing the pathway terms that are enriched in ammonia treatment. The color shades refer to the extent of up-regulation.

**Figure 11 genes-14-00795-f011:**
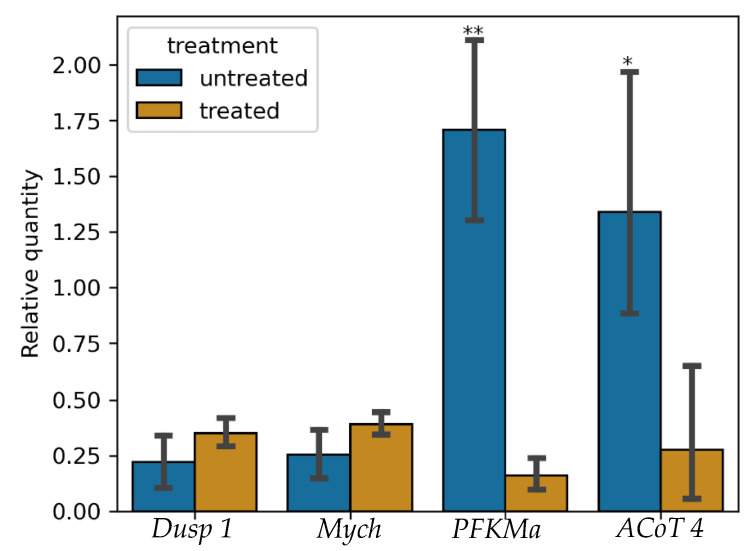
Transcript abundance of *DUSP 1*, *Mych*, *PFKMa*, and *AcoT4* when Nile tilapia was stressed with 0.12 mg/L of ammonia and stocking density of 70 fish per tank measured by RT-qPCR (one-way ANOVA; * indicates *p* ≤ 0.05 while ** indicates *p* ≤ 0.001).

**Table 1 genes-14-00795-t001:** Primer sequences for selected genes used during the RT-qPCR validation of the RNA-seq results.

Gene Name	Primer Sequence	Annealing Temp.	Amplicon Size	GC Content%
dusp1_F	ACTTGAACACATCGTCCCCAAC	61.1 °C	181 bp	50.0
dusp1_R	TGTGCCGTCCTTTTTCACTTGG	61.8 °C		50.0
mych_F	AACTGTTTCATGCGCTTGCG	60.7 °C	190 bp	50.0
mych_R	TTTCCTGGTTTGCAGTCTGTGG	61.3 °C		50.0
pfkma_F	TGCAATTAAGGCCAAGCACCAC	62.2 °C	196 bp	50.0
pfkma_R	GCATGGTGTTGTAAAGGCTCACC	62.5 °C		52.2
AcoT4_F	TTGAGGCAGGTGGTGAAGACAAG	62.9 °C	185 bp	52.2
AcoT4_R	TGTGCAACATAACAGGCAGCTG	62.0 °C		50.0
GAPDH_F	TGGCATTGCACTCAACGACAAC	62.5 °C	182 bp	50.0
GAPDH_R	GTGCAGCAAACAAGCTTTGGTC	61.6 °C		50.0

**Table 2 genes-14-00795-t002:** Mean length and weight relative condition factors and specific growth rates for Nile tilapia subjected to ammonia treatment.

Ammonia			Length–Weight Relationship			
Concentration	Length	Weight	a	b	K_n_	SGR (%)
Control (0 Mg/L)	13.4 ± 1.85	56.32 ± 22.28	0.01397	3.12202	1.17	1.81341
0.4 Mg/L	13.2 ± 1.67	52.56 ± 20.92	0.01415	3.12788	1.13	1.78352
0.8 Mg/L	13.1 ± 1.33	51.18 ± 17.88	0.01418	3.1277	1.11	1.64517
1.2 Mg/L	13.0 ± 1.80	46.37 ± 17.86	0.01437	3.11417	1.08	1.56446
1.6 Mg/L	12.8 ± 1.39	40.93 ± 12.30	0.01484	3.08808	1.02	1.5114
2.0 Mg/L	12.7 ± 1.34	38.70 ± 10.00	0.0154	3.07442	0.99	1.3249
2.4 Mg/L	12.5 ± 1.14	35.09 ± 8.87	0.01586	3.06278	0.93	1.22908

**Table 3 genes-14-00795-t003:** Mean length and weight relative condition factors and specific growth rates for Nile tilapia subjected to stocking density treatment.

Density			Length–Weight Relationship			
Fish/Tank	Length	Weight	a	B	K_n_	SGR (%)
Control (20 Fish/Tank)	14.1 ± 1.95	52.05 ± 21.34	0.012961	3.0635	1.17	1.567213
25 Fish/Tank	14.0 ± 1.86	48.54 ± 20.46	0.013067	3.0524	1.14	1.540296
40 Fish/Tank	13.8 ± 1.66	46.44 ± 17.64	0.013103	3.0601	1.11	1.329046
55 Fish/Tank	13.7 ± 1.65	41.62 ± 10.88	0.013156	3.0578	1.07	1.065474
70 Fish/Tank	13.4 ± 1.19	37.10 ± 8.46	0.013323	3.0490	1.01	0.905001
85 Fish/Tank	13.1 ± 1.17	34.34 ± 8.15	0.013327	3.0711	0.95	0.893896
100 Fish/Tank	13.1 ± 1.21	32.20 ± 7.43	0.013445	3.0627	0.91	0.812368

## Data Availability

The primary data used to support the findings of this study are available from the corresponding author upon request, however part of the data associated with this work including the whole genome sequence has been deposited in NCBI database Accession number JANFCW000000000. Other sequences may be subsequently be deposited in NCBI.

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
