# Peer review of "Whole Transcriptome Analysis of Differentially Expressed Genes in Cultured Nile Tilapia (O. niloticus) Subjected to Chronic Stress Reveals Signaling Pathways Associated with Depressed Growth"

_genes, 2023, doi:10.3390/genes14040795_

Round 1

Reviewer 1 Report

The authors of this manuscript mainly used different ammonia concentrations and stocking densities to observe gene expression profiles involved in chronic stress in the Nile tilapia transcriptome profiles. After reading the abstract a few times, I feel that the different ammonia concentrations and stocking densities are confusing and need to provide more specific goals. To me, although the author's strategy is to provide the most data, this is the main problem in this study. For example, if differentially expressed genes can be confirmed from the same set of target data sets, it will be very helpful to this effort and will have more potential. In fact, the overall research goals need to be broadened and made as unspecific as possible.

Special comments:

1)      The author's description of the experimental sample collection in the materials and methods is not detailed enough? For example, three animals are collected in each group, how many animals are collected in total, how many gene banks are finally established, and so on. It is recommended to add.

2)      Is there any damage or death to the fish during the stress test? If so, the authors should calculate the mortality rate for each treatment group. If the difference in mortality rate is too large, there will be problems of unobjective growth due to different stocking densities when calculating the growth performance data of fish in each group.

3)      Background information on creating transcriptome-sequenced samples is severely lacking. For example: What kind of ammonia concentrations or what kind of stocking densities groups are used as analysis representatives?

4)      From the Volcano plot in Figure 2, it can be found that the author has specially marked four genes regulated by myosin light chain 13, myosin 7, acyl-coenzyme thioesterase 4, and kchab1b. However, what is confusing is that the DUSP 1, Mych, PFKMA and Acyl coA Thioesterase 4 genes selected for real-time quantitative PCR verification (the reason is not clear), in addition to gene differences (the gene names are not consistent), actual validation of gene expression has also been inconsistent. This part really needs to ask the author to elaborate.

General comments:

1)      L170, L181: Inconsistent number of trial replicates.

2)      L253: The writing content of HISAT2 database is unclear.

3)      Table 1: It is suggested: (1) Changing GAD to GAPDH will be clearer. (2) Delete the "," symbol after 182nt and 62.0 C. (3) The temperature unit in the "Annealing temp." column should be changed to "°C" instead of "C". (4) Changed "nt" to "bp" in the column of "Amplicon size".

4)      Figure 2: In the three volcano maps of A, B, and C, the colors of the up-regulated and down-regulated expressions should be unified.

5)      Figure 3 and Figure 4: The number of genes shown and marked in up-regulation and down-regulation is just opposite to the number described in the content of L371-374. It is strongly recommended that the author must reconfirm the original data and correct the mistakes.

6)      Figure 5: X and Y axis titles are not clearly labeled. What does the Y-axis value represent? Does not match the number indicated above the bar graph.

7)      Figure 6: It's really hard to understand the "significantly enriched pathways" from the histogram of the GO pathway analysis.

8)      L441:(Figure 7 a). On…

9)      Figures 8, 9, and 10: Suggested revisions: (1) The results of up-regulated genes presented are not consistent with the description in the text of L455-461. (2) The number of DEGs and the up-down regulation information should be marked. (3) Why only the KEGG pathway of the ammonia stress group (without stocking densities) is presented. (4) Heat-Map of KEGG should be supplemented.

10)  Figure 11: The author did not clearly state that “Treated” refers to the verification results of the test under the stress treatment of ammonia (0-2.4 mg/L) or stocking density (25-100 fish per tank)? In addition, significant differences should be marked in the real-time quantitative PCR graph.

11)  Figure 11: The first letter of the caption should be capitalized.

12)  L495-496: Add closing brackets “)”.

13)  L526-527: "ventilation rates compared to" in the sentence does not use italics.

Reviewer 2 Report

The authors study the gene expression profiles associated with chronic stress in cultured Nile tilapia reared for 70 days at different ammonia concentrations and stocking densities. The research deals with the mechanism of the effect of chronic stress on weight loss in tilapia fish, which is beneficial to aquaculture systems.  The experimental design is generally sound and the data are adequate, which has some research value and practical significance. But, before publication, a few changes are required.

1-     The introduction is too long. It needs to be simplified and be more representative of the work. e.g. data on other species than tilapia is not required. Moreover, the introduction should be ended with the aims of the work which facilitate the understanding of the following data.

2-     Line 177, the first (control) group was maintained in natural water without the addition of ammonia throughout the growth period. Did the authors actually measure the ammonia concentration in this group? Please provide the concentration of ammonia in natural water in the control group.

3-     Line 179, Six groups of 25 fish per tank were maintained at different concentrations of unionized ammonia. Give in detail the source, and mode of application of UIA.

4-     Footnotes of the results tables should be clearer and more illustrative.

5-     Did any mortality appear during the experiment?

6-     Line 526. Problem with the format.

7-      Moderate English changes are required in the manuscript. Has many long sentences

Round 2

Reviewer 1 Report

Figure 2 and 11: 

(1) Real-time quantitative PCR verification (Figure 11) shows that the Acyl coA Thioesterase gene is down-regulated, why is it inconsistent with the up-regulation result of transcript sequencing (Figure 2A)? The reason should be explained clearly.

(2) Is myl13 in Figure 2A the same gene as mych in Figure 11? If so, all gene names should be presented consistently in the manuscript content (including figures).

Figure 11:

The author did not respond to questions at all. My question is: the author should clarify that "Treated" refers to the set of ammonia stress treatments in the range (0-2.4 mg/L) of 0 (control), 0.4 (treated 1), 0.8 (treated 2), 1.2 (treated 3), 1.6 (treated 4), 2.0 (treated 5), 2.4  (treated 6) mg/L, or the set of stocking densities tested in the range (20-100 fish/tank) of 20 (control), 25 (treated 7), 40 (treated 8), 55 (treated 9), 70 (treated 10), 85 (treated 11), 100 (treated 12) fish/tank validation results?

In addition, the authors did not mark significant differences in the real-time quantitative PCR plots. Strongly suggest what significant difference the authors should mark between the two groups

Reviewer 2 Report

I thank the authors for their efforts in answering the inquiries and amending the manuscript

Round 3

Reviewer 1 Report

I have no comments or any suggestions.